# Preventing antimalarial drug resistance with triple artemisinin-based combination therapies

Tran Dang Nguyen[1,4], Bo Gao[2,4], Chanaki Amaratunga[2,3], Mehul Dhorda[2,3], Thu Nguyen-Anh Tran[1], Nicholas J. White[2,3], Arjen M. Dondorp[2,3], Maciej F. Boni[1,2,4] ✉ & Ricardo Aguas[2,3,4] ✉

Increasing levels of artemisinin and partner drug resistance threaten malaria control and elimination globally. Triple artemisinin-based combination therapies (TACTs) which combine artemisinin derivatives with two partner drugs are efficacious and well tolerated in clinical trials, including in areas of multidrug-resistant malaria. Whether early TACT adoption could delay the emergence and spread of antimalarial drug resistance is a question of vital importance. Using two independent individual-based models of *Plasmodium falciparum* epidemiology and evolution, we evaluated whether introduction of either artesunate-mefloquine-piperaquine or artemether-lumefantrine-amodiaquine resulted in lower long-term artemisinin-resistance levels and treatment failure rates compared with continued ACT use. We show that introduction of TACTs could significantly delay the emergence and spread of artemisinin resistance and treatment failure, extending the useful therapeutic life of current antimalarial drugs, and improving the chances of malaria elimination. We conclude that immediate introduction of TACTs should be considered by policy makers in areas of emerging artemisinin resistance.

The introduction of artemisinin-based combination therapies (ACTs) into routine clinical use for uncomplicated *Plasmodium falciparum* malaria has saved millions of lives over the past two decades[1]. These drugs are the mainstay of antimalarial treatment and remain highly effective in most malaria-endemic regions[2]. However, resistance to artemisinins has emerged in several malaria-endemic regions, first in the Greater Mekong Subregion (GMS) of Southeast Asia[3–5], and subsequently in South America[6], Papua New Guinea[7], and most recently in Eastern Africa[8,9]. In the GMS, artemisinin resistance was compounded by ACT partner drug resistance, causing ACT treatment failure[10,11]. Increasing drug resistance threatens global malaria control and elimination efforts. Artemisinin partial resistance, caused by mutations in the *pfkelch13* gene[12,13], results in slower parasite clearance, increased

rates of treatment failure, increased transmissibility, and reduced protection of the partner drugs from resistance emergence and spread. The recent emergence of artemisinin resistance in East Africa is particularly concerning as the previous reduction in the malaria burden has stalled in many parts of the continent. Malaria mortality, which fell substantially between 2000 and 2015, is estimated to have plateaued over the past 8 years[2]. Worsening drug resistance will result in an increasing death toll, with most of these preventable deaths occurring in African children.

The current high efficacy of antimalarial treatments must be maintained if a lethal reversal in malaria trends is to be avoided. New antimalarial drugs are promised[14], but even if their development continues successfully, they may not become available for years and will

---

[1]Center for Infectious Disease Dynamics, Pennsylvania State University, University Park, PA, USA. [2]Centre for Tropical Medicine and Global Health, Nuffield Department of Medicine, University of Oxford, Oxford, UK. [3]Mahidol-Oxford Tropical Medicine Research Unit, Faculty of Tropical Medicine, Mahidol University, Bangkok, Thailand. [4]These authors contributed equally: Tran Dang Nguyen, Bo Gao, Maciej F Boni, Ricardo Aguas. ✉e-mail: mfb9@psu.edu; ricardo@tropmedres.ac

be more expensive[15]. Multiple first-line therapies (MFT), which mathematical modeling suggests can slow drug resistance spread[16,17], have become policy in more than a dozen endemic countries[2]. However, there is as yet no field evidence demonstrating the success of MFT at slowing or delaying resistance evolution. A potential alternative solution, based on the same biological principles of combining two different slowly eliminated antimalarial drugs with an artemisinin derivative, are triple artemisinin-based combinations (TACTs), which should provide protection against partner drug resistance and therefore preserve high treatment efficacy, given the extreme rarity of parasites acquiring mutations conferring resistance against both partner drugs over a short period of time. Randomized clinical trials in Asia with dihydroartemisinin-piperaquine-mefloquine (DHA-PPQ-MQ) and artemether-lumefantrine-amodiaquine (ALAQ) have shown that these combinations are well-tolerated, safe, and effective, including in areas with multidrug-resistant falciparum malaria[18,19]. Dose-optimized TACTs are now being tested in large trials in African and Asian countries (clinicaltrials.gov identifiers NCT03923725 and NCT03939104, respectively). However, the long-term evolutionary benefits of TACT deployment cannot be assessed with clinical trials. Here, we use a consensus mathematical modeling approach to project the potential long-term evolutionary dynamics and clinical treatment outcomes of TACT deployment in different malaria epidemiological settings. The results can inform proactive policies aiming to contain the emergence and spread of antimalarial drug resistance.

## Results

The models predicts that replacing ACTs with triple artemisinin-based combination therapies (TACTs) substantially slows the emergence and evolution of artemisinin-resistant alleles and restores the clinical efficacy of first-line therapy (Fig. 1). Notably at model year zero, under a scenario where DHA-PPQ had been used for the previous 15 years, the mutant *pfkelch13* 580Y alleles were present in both models; at 0.013 (IQR: 0.006–0.043) allele frequency in the PSU model and 0.080 (IQR: 0.065–0.096) allele frequency in the MORU model. Under continued DHA-PPQ use, the allele frequency of *pfkelch13* 580Y increased to a median frequency of 0.884 (IQR: 0.725–0.968; PSU) or 0.571 (IQR: 0.335–0.769; MORU) after 10 years. In both models, under continued ACT use, ten years was sufficient time for artemisinin-resistant genotypes to become established (Fig. 1 and Supplementary Figs. 2–7). If ASMQ-PPQ was deployed at year zero replacing DHA-PPQ, *pfkelch13* 580Y alleles were projected to reach frequencies of 0.056 (IQR: 0.014–0.226; PSU) or 0.155 (IQR: 0.112–0.224; MORU) after 10 years. Under ALAQ deployment, 580Y emergence was slower under the PSU model, with predicted median allele frequency of 0.015 after 10 years (IQR: 0.003–0.054), but similar under the MORU model (median = 0.162; IQR: 0.116–0.213). The relative benefit of ALAQ over ASMQ-PPQ was sensitive to the model used and scenario examined (Fig. 2). The risk of an infection carrying the *pfkelch13* 580Y allele after only 2 years of ALAQ (TACT) deployment was significantly lower than that with continued ACT use, with mean relative risks of 0.49 (95% CI: 0.36-0.66; PSU) and 0.71 (95% CI: 0.55-0.91; MORU). With ASMQ-PPQ (TACT) deployment, the relative risks were 0.49 (95% CI: 0.35–0.68; PSU) and 0.93 (95% CI: 0.73–1.19; MORU). The corresponding relative risks at year 10 were 0.19 (95% CI: 0.16–0.22; PSU) and 0.33 (95% CI: 0.29–0.38; MORU) for ASMQ-PPQ and 0.10 (95% CI: 0.08–0.12; PSU) and 0.31 (95% CI: 0.27–0.36; MORU) for ALAQ.

Both models agreed that the lower *pfkelch13* 580Y frequencies resulting from TACT deployment would result in lower population-wide treatment failure rates. For our standard evaluation scenario (defined as 1% PfPR, 50% coverage, and baseline DHA-PPQ use), the projected treatment failure rates (measured at 28 days post-treatment) after 10 years of continued ACT use were 52.0% (IQR: 47.1%–54.4%) for the PSU model and 20.6% (IQR: 12.6%–29.1%) for the MORU model.

With the introduction of TACTs (either ASMQ-PPQ or ALAQ), the upper quartile of treatment failure rates was projected to stay below 15% (for both models and both TACTs) for the entire ten-year period. This resulted from (i) the increased efficacy of TACTs relative to ACTs (Supplementary Fig. 1) and (ii) the much slower emergence and spread of *pfkelch13* 580Y alleles (Fig. 2; Supplementary Figs. 2–7). Despite different model predictions for the speed of spread of artemisinin-resistant alleles, both models projected that TACT introduction (at 1% prevalence and 50% treatment coverage) would lead to a >74% reduction in treatment failure rates for countries currently using baseline DHA-PPQ, >34% treatment failure rate reduction in scenarios with ASAQ as baseline, and >17% reductions when AL is the baseline treatment (Fig. 3; Supplementary Figs. 8–13).

In 52 out of 54 scenarios examined (27 per model) the 10-year treatment failure rates were lower when TACTs were introduced compared to continued ACT deployment (Fig. 3; nearly all Mann–Whitney p values < 10⁻⁴; see Supplementary Figs. 20, 21). In the other two scenarios, both in a low-transmission setting (75% treatment coverage, 0.1% prevalence, baseline AL use, PSU model; and 25% treatment coverage, 0.1% prevalence, baseline DHA-PPQ use, MORU model) the simulations projected median prevalence levels below 0.05% after 10 years with median treatment failure rates of zero both for continued ACT use and switch to TACT. Thus, no scenarios showed any advantages of ACT deployment over TACT deployment–Supplementary Figs. 22–28.

### Preventing drug resistance

The short-term benefits of a switch to TACTs (Fig. 1) are likely to be critically dependent on the initial conditions at year zero (the year of TACT introduction) and the expected evolutionary trajectories of drug resistance allele frequencies. Treatment failure and allele frequency dynamics are highly nonlinear and grow exponentially in their early phases (Supplementary Figs. 29–34), suggesting that the timing of TACT introduction (i.e., as early as possible) could be critical to its long- and short-term success.

Following the standard scenarios (Fig. 1), switching to TACTs led to median artemisinin resistance allele frequencies ranging from 0.01 to 0.17 (across the two models and two TACTs) ten years after TACT deployment. If TACT introduction were delayed by three years, the median projected 10-year *pfkelch13* 580Y allele frequency ranged from 0.08 to 0.22 and shifted to 0.24 to 0.45 if TACT introduction were delayed by 5 years. This behavior is consistent between models across different prevalence settings although the magnitude of the effect over ten years differed between the models (Fig. 4A, B). In very low prevalence settings delays in TACT implementation could preclude elimination (Fig. 5).

It should be noted that while we are presenting 580Y frequency as the signature genetic marker of artemisinin resistance and treatment failure, there are differences in how selection operates across resistance loci. For example, piperaquine resistance and artemisinin resistance are co-selected strongly in a scenario where DHA-PPQ is the recommended ACT (Supplementary Fig. 34). The reason is that the treatment failure rate of DHA-PPQ on the double-resistant genotype is very high. Continued AL deployment, on the other hand, facilitates mdr1 second copy selection (Supplementary Fig 35), which is further enhanced if changing to ASMQ-PPQ but mitigated if changing to ALAQ. AL is the only regimen to select strongly for N86, although this can be reversed after a policy change to TACTs (Supplementary Fig 35). Under high selection pressure scenarios, ASMQ-PPQ consistently selects for a higher frequency of drug resistance-related alleles relative to ALAQ (Supplementary Figs. 34–36). Continued use of ASAQ is predicted to result in high long-term 580Y frequencies but barely noticeable increases in all other resistance-related allele frequencies.

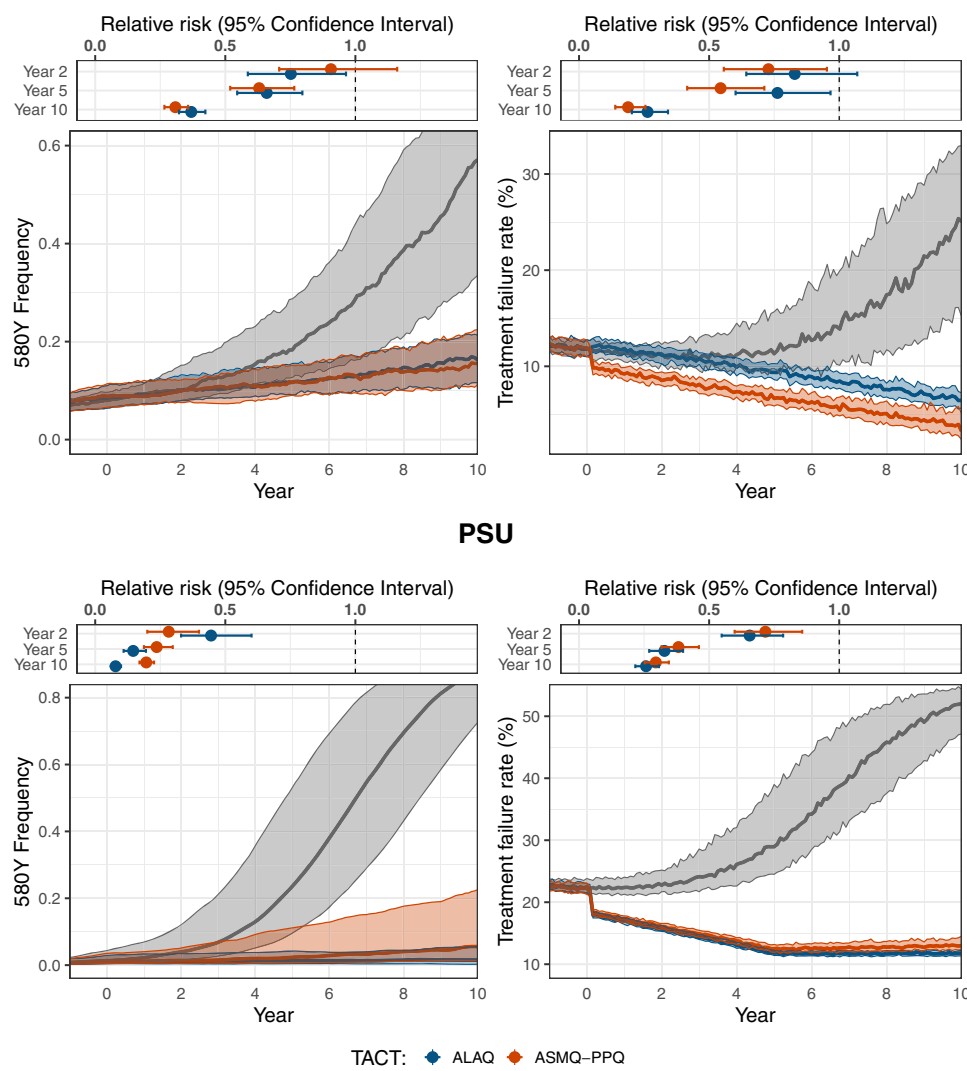

**Fig. 1 | TACT longitudinal impact on 580Y frequency and treatment failure rate.** Comparison of artemisinin-combination therapy (ACT) and triple artemisinin-combination therapy (TACT) deployment at 1% *Plasmodium falciparum* parasite ratio (PfPR) and 50% treatment coverage, with dihydroartemisinin-piperaquine used as the baseline ACT before TACTs are deployed at year zero; gray lines (medians from 100 simulations) show the evolution of the *pfkelch13* 580Y allele or treatment failure rates under continued dihydroartemisinin-piperaquine use. Red lines show how these processes are slowed down by deployment of artesunate-mefloquine-piperaquine (ASMQ-PPQ). Blue lines show how these processes are slowed down by deployment of artemether-lumefantrine-amodiaquine (ALAQ). All shaded areas show interquartile ranges. Panels above each graph show an individual's relative risk of 580Y infection (under TACT deployment versus ACT deployment) after 2, 5, and 10 years of deployment; bars show 95% confidence intervals and dots indicate the median, assuming a sample size of *n* = 1000. Results for baseline artesunate-amodiaquine and artemether-lumefantrine use are shown in Supplementary Figs. 52 and 53.

## Pre-existing triple resistance mechanisms

The major long-term benefit of combination therapy, that multidrug resistance emerges much later under combination therapy than single-drug resistance does under monotherapy, relies on multidrug-resistant genotypes being absent from the population at the time of deployment. To evaluate the risk posed by pre-existing multidrug resistance, we explored a range of starting ASMQ-PPQ triple-resistant genotype frequencies in a PfPR setting of 0.1%. If the triple-resistant genotype's frequency was lower than 0.01 at the time of TACT deployment, the models showed no major effect on the emergence of parasites resistant to all 3 drugs (Fig. 6). However, a marked increase in the risk of triple drug-resistant mutant spread was predicted when the initial frequency of these mutants was between 0.01 to 0.05 (Fig. 6A). For example, when triple-mutant frequency was set to 0.04 at year zero, natural selection operated efficiently (i.e., no stochastic disappearance due to genetic drift) to produce median frequencies of

0.35 (IQR: 0.11–0.62; PSU) and 0.25 (IQR: 0.00–0.55; MORU) after 10 years. This selection is enabled by the model-calculated 71% to 73% treatment efficacy of ASMQ-PPQ for infections with the triple-resistant genotype (see PKPD section of the methods).

Following the selection of triple-resistant parasites, the probability of reaching elimination within ten years in a 0.1% prevalence setting declined substantially. For triple-resistant frequencies below 0.01 at the start of TACT deployment, 10-year elimination probabilities were ~80% (PSU) and 90% (MORU). However, a starting triple-resistance frequency of 0.04 led to 43% (PSU) and 30% (MORU) probabilities of elimination, with a steep drop to a <5% probability for falciparum malaria elimination with initial triple-resistant frequencies above 0.20 (Fig. 6B). If prevalence is higher than 0.1%, or if treatment coverage is higher than 50%, natural selection of the triple-resistant is stronger and less susceptible to the action of random genetic drift at small population sizes. Under these conditions, a triple-resistant

## 580Y Frequency

*Expected 580Y frequency at year 10 for 50% treatment coverage*

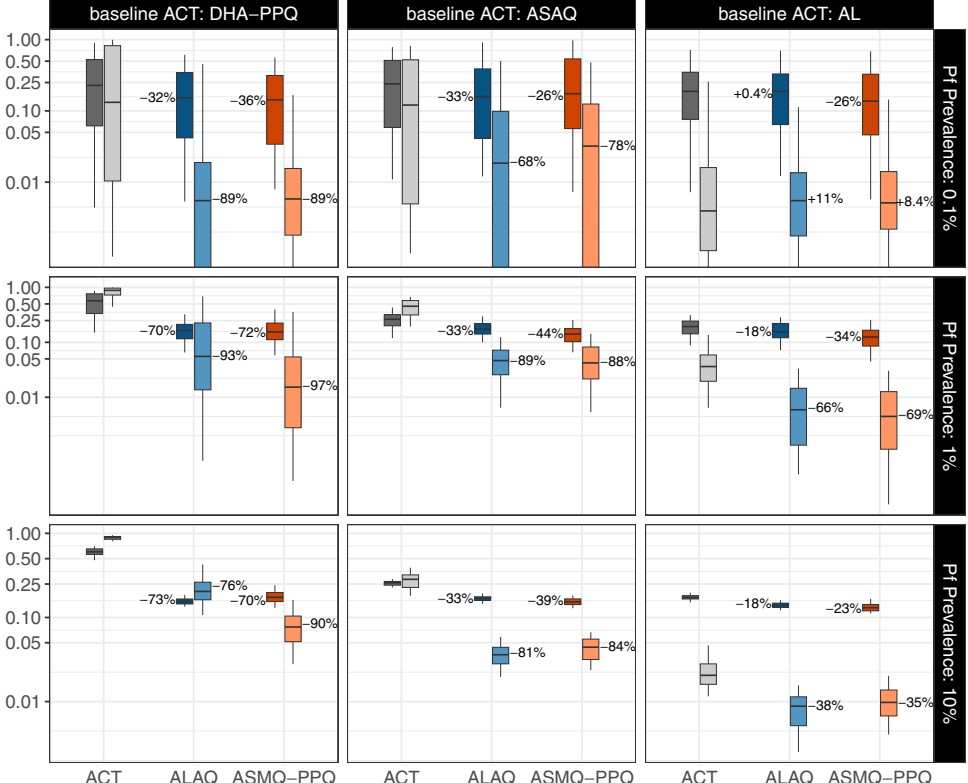

**Fig. 2 | TACT long-term impact on 580Y frequency.** *Pfkelch13* 580Y allele frequencies after 10 years of triple artemisinin-based combination therapy (TACT) deployment or artemisinin-based combination therapy (ACT) deployment, in three baseline scenarios of ACT use (columns) and three prevalence settings (rows). Treatment coverage is 50%; Supplementary Figs. 22 and 23 show results for treatment coverages of 25% and 75%. The gray boxplots in each panel show 580Y allele frequencies (*y* axes) 10 years later under a status quo ACT policy. The blue (artemether-lumefantrine-amodiaquine) and red (artesunate-mefloquine-piperaquine) boxplots show 580Y frequencies after 10 years of a TACT policy. Boxplot pairs have Mahidol–Oxford research unit model results on the left and Penn State University model results on the right. The percent reduction in median 580Y allele frequency from ACT to TACT deployment is shown next to the median line of each TACT boxplot. All boxplots summarize 100 simulations, with boxes displaying interquartile (IQR) range and whiskers extending to 1.5 times the IQR.

genotype frequency >0.01 at the start of TACT deployment predictably led to the rapid selection of the triple-resistant and near-term increases in treatment failure (Supplementary Figs. 34–36).

## Discussion

The efficacy of TACT for the treatment of drug-resistant and drug-sensitive malaria can be measured in randomized controlled trials. However, their long-term population-wide impact on prevalence, treatment failures, and drug resistance can only be predicted with mathematical models that relate malaria case management to malaria epidemiology. Consistent with expectations from evolutionary biology, the two independently validated models[16,20] of malaria epidemiology and drug resistance evolution evaluated here found that triple therapies (TACTs) are more effective than double therapies (ACTs) at delaying the spread of drug resistance. Safeguarding the long-term efficacy of antimalarial therapy is essential for reaching the ultimate goal of malaria elimination. The two models agreed that the main driver of loss of ACT therapeutic efficacy is the emergence and spread of artemisinin resistance (Supplementary Figs. 37–39). We showed that a switch to TACTs as first-line therapy can delay the selection of *pfkelch13* 580Y mutant parasite lineages substantially, even when these are already present at low allele frequency, because TACT deployment inhibits their spread when both partner drug-resistant mutations are absent. In evolution, the acquisition of mutations or additional gene copies in the same genome occurs nonlinearly in time: if it takes *y* years for a *P. falciparum* lineage to acquire a particular set of mutations it will

take $y^2$ years to acquire twice as many. An exception to this rule is predicted only under very high rates of recombination[21,22].

Comparing the two TACT options evaluated, the persistence of artemisinin-resistant alleles is predicted to be disrupted most, and thus resistance delayed for longer, by the deployment of ALAQ. This is because of the inverse relationship between resistance to amodiaquine and lumefantrine[23,24] Our results suggest that in unusual low to moderate drug selection pressure settings, i.e., low treatment coverage (≤50%) and low *Pf* prevalence (≤1%), the 10-year benefits of ALAQ compared to ASMQ-PPQ are not substantially different. This results from (i) the higher efficacy of PPQ as compared to the other partner drugs (Supplementary Fig. 1) and (ii) the higher efficacy of ASMQ-PPQ on its single-resistant and double-resistant mutants. In contrast, in areas of higher transmission intensity with good treatment coverage (≥75%), ALAQ is superior in preventing the spread of artemisinin resistance and conserving therapeutic efficacy (Supplementary Figs. 29–31). Notably, all TACT policies evaluated here are non-adaptive, i.e., they do not use concurrent surveillance information to adjust treatment guidelines. An adaptive TACT strategy, based on real-time resistance data, would lead to an even more favorable TACT assessment than the one discussed here.

It is generally agreed that investing in preventing the emergence of resistance to any anti-infective therapy is preferable to reacting to an epidemic of drug resistance[25-27]. It is cost-beneficial, and it results in less morbidity and mortality. Where resistance arises very rapidly within an individual, such as in tuberculosis or HIV infection, triple or quadruple therapies were soon adopted as the standard of care since

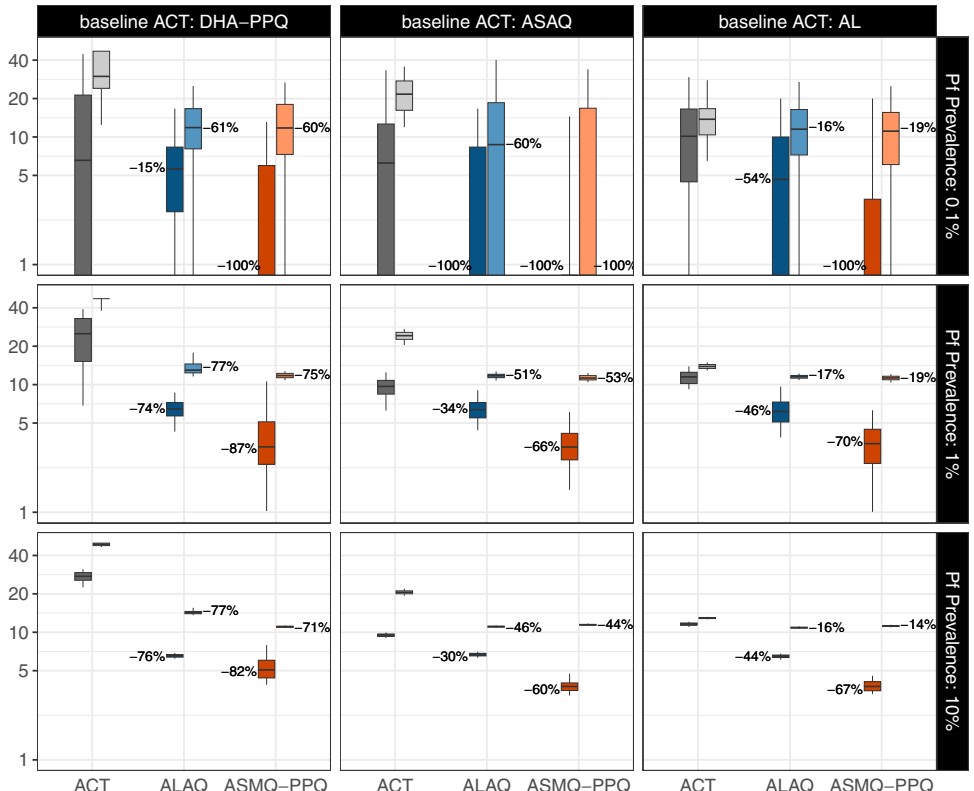

**Fig. 3 | TACT long-term impact on treatment failure rate.** Treatment failure rates after 10 years of triple artemisinin-based combination therapy (TACT) deployment or artemisinin-based combination therapy (ACT) deployment, in three baseline scenarios of ACT use (columns) and three prevalence settings (rows). Treatment coverage is 50%; Supplementary Figs. 24 and 25 show results for treatment coverages of 25% and 75%. The gray boxplots in each panel show the treatment failure rates (y axes, %) 10 years later, under a status quo ACT policy. The blue (artemether-lumefantrine-amodiaquine) and red (artesunate-mefloquine-piperaquine) boxplots show treatment failure rate outcomes after 10 years of a TACT policy. Boxplot pairs have Mahidol−Oxford research unit model results on the left and Penn State University model results on the right. The percent reduction in median treatment failure rates from ACT to TACT is shown next to the median line of each TACT boxplot. All boxplots summarize 100 simulations, with boxes displaying interquartile (IQR) range and whiskers extending to 1.5 times the IQR.

the value of resistance prevention to individual patient outcomes was obvious. Antimalarial drug resistance is more insidious. Over the past seventy years, antimalarial drug resistance has killed millions, mainly African children[28]. Yet, public health response has generally been slow[29], with treatment policies only changing once resistance had become firmly established and the antimalarial drugs were either failing or completely ineffective. If TACTs were adopted now in Africa, where artemisinin resistance is still mostly absent, they could have a large impact on the course of artemisinin resistance evolution, and thereby avoid preventable malaria morbidity and mortality. In areas such as Rwanda and Uganda, where several *pfkelch13* mutants have increased to >0.20 allele frequency regionally, TACTs could have a clear and immediate impact[8,9]. To quantify how the benefits of delaying artemisinin and partner drug resistance can be compromised by decision-making inertia, we evaluated scenarios where TACTs were introduced with delays of one to five years. The two independent models agree that delays in TACT deployment will result in higher long-term *pfkelch13* 580Y frequency and more treatment failures (Fig. 4), increasing morbidity and compromising the likelihood of elimination in low-transmission settings (Fig. 5). Immediate but gradual TACT adoption will have a similar impact on *pfkelch13* 580Y frequency and treatment failure rates (Supplementary Fig. 54).

Use of individual antimalarial drugs at low doses and/or with poor adherence, or with low-quality medicines, enables the emergence of resistance. In Southeast Asia, *P. falciparum* has acquired resistance mechanisms to all the antimalarial drugs that have been deployed,

including mefloquine and piperaquine. We, therefore, investigated to what extent triple-resistant mutants undermine malaria elimination efforts and resistance containment strategies. The potential for selection of a pre-existing multidrug-resistant genotype is the Achilles' heel of any combination therapy. Our models predicted that even a 0.01 multidrug resistance genotype frequency affecting all three components of the ASMQ-PPQ TACT results in the efficient and predictable natural selection of this triple-resistant mutant. Above this frequency threshold, this multidrug-resistant genotype will no longer be susceptible to stochastic loss through bottlenecking or random genetic drift, and natural selection will act efficiently to bring the multi-resistant genotype up to high frequencies. Continuous molecular surveillance is thus particularly important for ASMQ-PPQ deployment.

## Limitations

Our two independent models make different assumptions on mechanisms of immunity, symptoms presentation (Supplementary Table 1), and within-host evolution. This is important for TACT evaluation because the selection pressure exerted by TACTs will differ by genotype, but the impact of changes in fitness conferred by each genotype in any drug resistance model is dependent on treatment coverage[30] and symptoms presentation[17]. If these have been parameterized incorrectly, the true patterns of selection may be different than modeled here. To address this issue, we calibrated model parameters to force the two models to reach 0.01 artemisinin-resistant allele frequency at the same time under a specified set of conditions.

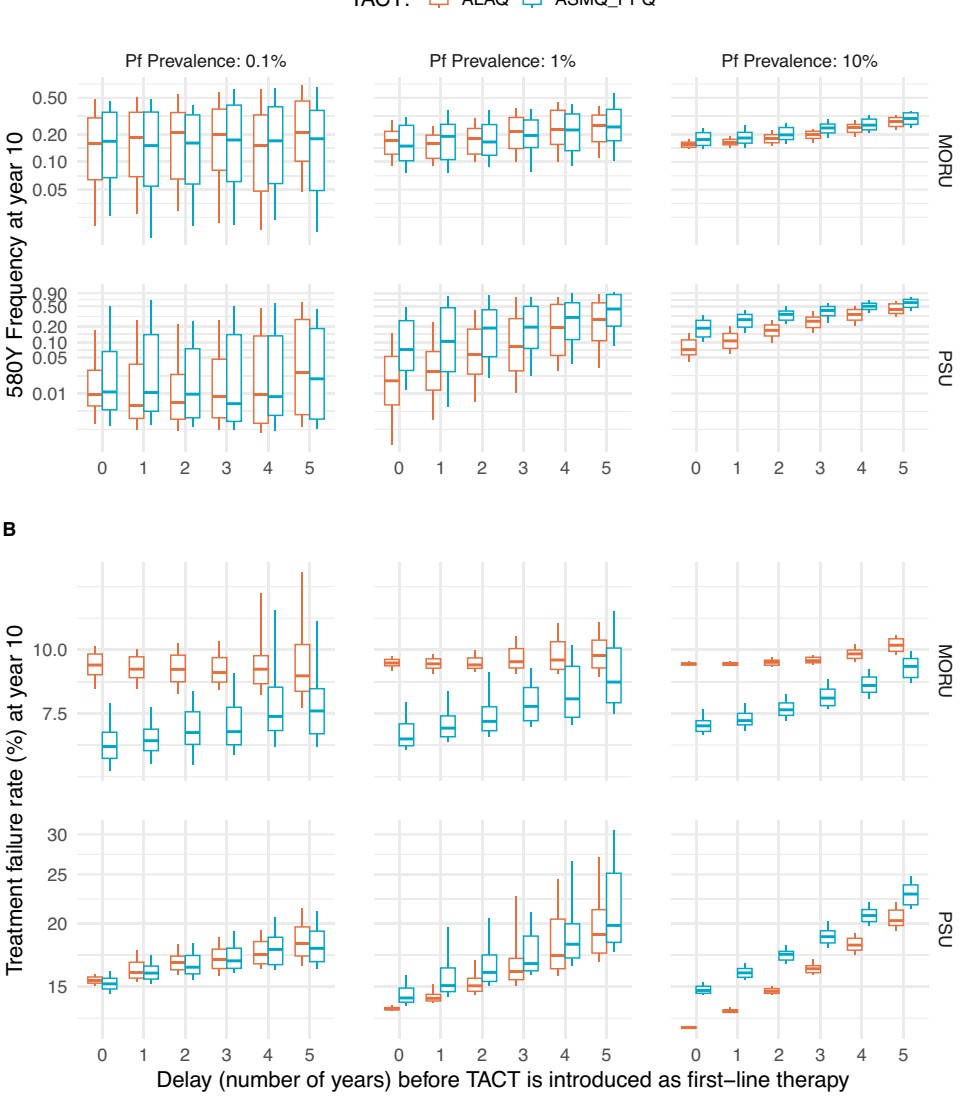

**Fig. 4 | Increases in artemisinin resistance frequency due to late adoption of TACTs.** Results are shown for the Mahidol–Oxford research unit model (top row) and the Penn State University model (bottom row), for three different prevalence levels (columns), fixing treatment coverage at 50%. In each panel, the x-axis shows the number of years of delay before a triple artemisinin-based combination therapy (TACT) is introduced, and the y axis shows the artemisinin-resistant allele's frequency at year 10 (**A**) or treatment failure at year 10 (**B**). Boxplots show median and interquartile ranges (IQR) with whiskers extending to 1.5 times the IQR.

However, selection pressures across models do differ for the artemisinin-resistant 580Y allele frequencies above 0.01. Together with the dynamic use of four or more different drugs over time, this renders calibrating both models to the exact same epidemiological scenario impossible. This results in slightly different initial starting allelic frequencies (particularly *pfkelch13* 580Y), as well as diverging treatment failure rates, at the start of each prospective scenario. Reassuringly, the benefits of TACT over ACT deployment are robust to these model differences (Figs. 1–3).

There is limited information on the direction of resistance selection under ALAQ treatment as AQ selects for a certain group of alleles (76 T, 86Y, Y184) while lumefantrine selects for their counterparts (K76, N86, 184 F). The strengths of these selective pressures are difficult to estimate[31] particularly as relative strengths of intra-host selection change over time because lumefantrine is eliminated from the blood more rapidly than amodiaquine. We assumed that triple-resistant genotypes to ALAQ cannot emerge, as collateral sensitivity at three separate loci ensures that evolution towards lumefantrine resistance increases

amodiaquine sensitivity[24]. However, this could be compromised by novel lumefantrine resistance mechanisms which are not conferred by *pfmdr1* and *pfcrt*[32]. Triple resistance to ASMQ-PPQ, however, can and does emerge in the model simulations, by successive acquisition of the *pfkelch13* 580Y allele (or a similar *pfkelch13* allele conferring delayed parasite clearance), additional *pfmdr1* gene copies, and acquisition of piperaquine-resistant genotypes comprising a certain genetic background with increased *plasmepsin-2,3* gene copy number and particular *pfcrt* mutations. In addition, the spread of artemisinin resistance could also be driven by additional factors, such as increased transmissibility of *pfkelch13* mutant infections. In this scenario, the resistance delaying effect of TACTs on artemisinin resistance spread would be less than predicted by our model, but the mutually protective effect between ACT partner drugs would remain the same.

For specific in-country evaluations, the geography, variation in prevalence, monthly case burden, and drug access should all be locally parameterized to provide detailed model forecasts of the potential benefits of TACT deployment.

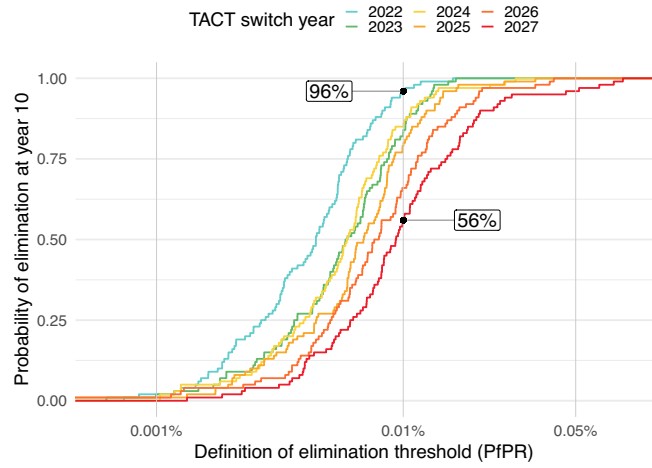

**Fig. 5 | Impact of TACT implementation delay on elimination prospects.**
Assuming a low-transmission setting with 0.1% *Plasmodium falciparum* parasite ratio (PfPR) where dihydroartemisinin-piperaquine is used as first-line therapy, we simulate a switch to artemether-lumefantrine-amodiaquine at year 0, assuming 50% treatment coverage. The probability of reaching elimination for each elimination threshold is given by the colored lines which indicate different delays in triple artemisinin-based combination therapy (TACT) adoption. Assuming no delay in switching to TACTs, the probability of attaining a PfPR of ≤0.01% was 96%. The corresponding probability is 56% when a 5-year delay in TACT implementation is imposed.

To conclude, here we show that two independent mathematical models agreed that deployment of TACTs will delay the emergence and spread of artemisinin and partner drug resistance substantially. This would secure effective antimalarial treatment for the next 5 to 10 years, a prerequisite for successful malaria elimination and control. Critically, the results advocate for immediate deployment of TACTs, as each year TACT deployment is delayed translates into diminishing long-term benefits. This emphasizes the importance of preventing antimalarial resistance rather than allowing resistance to establish and then trying to reverse it.

## Methods

### Individual-based malaria simulation models
Two published independently-built individual-based mathematical models of *Plasmodium falciparum* transmission and resistance evolution ("MORU"[20] and "PSU"[16,33]) were used to compare the population-wide benefits of deploying TACTs versus continued ACT use. These are microsimulation models run as daily time-step discrete-event simulations of individuals (humans) who can be infected with *Plasmodium falciparum* malaria and subsequently pass on their infection to other individuals in the simulation via mosquitoes (which are also explicitly modeled in the MORU model).

The parameterizations used here follow those in a previous consensus exercise[34], with three key differences: (1) population size was set to one million individuals, (2) private-market drug use was included, and (3) TACTs were included as antimalarial treatment. The inclusion of TACTs required the incorporation of their pharmacokinetic properties as well as pharmacodynamic parameters calibrated to treatment efficacies[31] consistent with those found in recent clinical trials[18] (Supplementary Fig. 1).

### Locus structure
The current state of the science in malaria individual-based modeling allows for only a limited number of loci to be included in an evolutionary analysis of drug resistance, as (1) there is no general genotype-phenotype relationship among all the known falciparum drug-resistant genotypes and their clinical/parasitological phenotypes and (2) there are

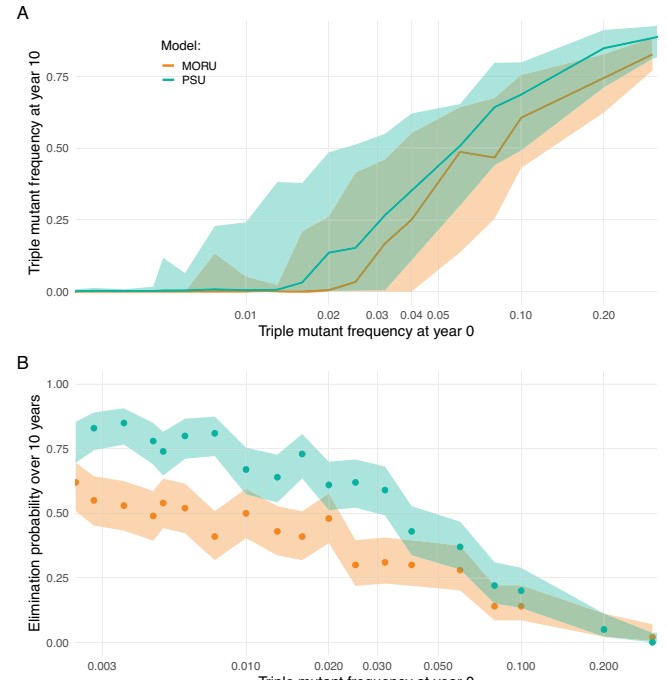

**Fig. 6 | Relationship between triple-mutant frequency and probability of elimination.** Assuming a low-transmission setting with 0.1% *Plasmodium falciparum* prevalence where dihydroartemisinin-piperaquine is used as first-line therapy, we simulate a switch to artesunate-mefloquine-piperaquine at year 0. At the time of first-line therapy switch, different frequencies of a pre-existing triple-mutant genotype conferring resistance against artemisinin, mefloquine, and piperaquine were imposed. For each initial frequency of the triple-resistant mutant (*x* axis) 100 simulations were run. After 10 years of simulation, the triple-mutant frequency was recorded, and the probability of elimination was calculated as the proportion of simulations reaching a prevalence lower than 0.01%. **A** Lines and the shaded bands show the median and interquartile range of obtained triple-mutant frequencies at year 10 for both models. **B** Lines and shaded bands show the model-predicted average and respective 95% confidence intervals across the 100 simulations for each value of triple-mutant frequency. Confidence intervals were generated using a one-sample proportions test without continuity correction.

computational limitations on the required size of recombination tables (necessary for modeling the sexual stage of falciparum reproduction) which are of dimension $2^n \times 2^n \times 2^n$ when $n$ loci are included. As in a previous publication[33], both models include six key drug resistance loci: *pfcrt* K76T, *pfmdr1* N86Y, *pfmdr1* Y184F, *pfkelch13* C580Y, copy number of *pfmdr1*. These are determinants of drug resistance to chloroquine, amodiaquine, lumefantrine, the artemisinins, and mefloquine. A generic piperaquine resistance haplotype/genotype is included, determined by several mutations in the *pfcrt* gene[35–37] and copy number of *Pfplasmepsin-2,3* genes. All parasites at model initialization carried wild-type *pfkelch13*, wild-type *pfmdr1* Y184, and single-copy *pfmdr1* and *pfplasmepsin2,3* (i.e., piperaquine sensitive). All four configurations of the K76T and N86Y loci (K76-N86, K76-86Y, 76T-N86, and 76T-86Y) were set to be equally represented in the parasite population, at the start of the simulation, to reflect diverse amodiaquine, lumefantrine, and chloroquine resistance profiles. The key *pfcrt* mutations that are known to be associated with piperaquine resistance[35] are not included in the model, and we assume a worst-case scenario in this analysis, i.e., that these background mutations are present bringing DHA-PPQ efficacy down to 41.5% on its double-resistant genotype[36,37] (580Y, double-copy plasmepsin). The 580 locus in *PfKelch13* can be viewed as a proxy for the evolution of 561H in Rwanda[8] as mutations in this locus have been associated with delayed parasite clearance[38]. We have used *pfplasmepsin2-3* amplification as the marker of piperaquine resistance,

although the primary causal mutations are in *pfcrt* downstream from the chloroquine resistance locus[37,39].

## Mutation and importation

Mutations and copy number increases/decreases can occur at all loci, and as in most modeling analyses in evolutionary epidemiology, the mutation process is modeled as the conversion of an entire within-host parasite population from one genotype to another. In other words, this process models the combined dynamics of mutation and within-host fixation of a new genotype. In the PSU model, this can only occur during a period of drug treatment and only from a less resistant genotype to a more resistant genotype (defined by higher treatment failure of the current therapy on that genotype). In the MORU model this same process is complemented by reverse mutations, where a very resistant genotype can mutate to a lesser resistant genotype in the absence of drug pressure. As this mutation-plus-fixation process is difficult to parameterize from field data, a previous calibration was used where both models' mutation processes are aligned to achieve 0.01 580Y allele frequency after 7.0 years of DHA-PPQ use at 40% coverage and 10% PfPR; this ensures that one model does not produce a larger number of de novo mutants than the other. Artemisinin-resistant 580Y mutants are also imported once every 100 days via a Poisson process.

**Drugs used.** A proportion of infectious mosquito bites in the model result in successful infection and symptoms (depending on a person's malaria history and current level of immunity) causing an individual to seek and receive antimalarial treatment with some probability. This probability (the 'treatment coverage') is set to 25%, 50%, and 75% in different sets of simulations, allowing us to examine scenarios with poor access to drugs or variation in individual choices to seek treatment or not. In our scenarios, the baseline ACTs available in the public sector–i.e., the recommended first-line therapies prior to TACT introduction–are dihydroartemisinin-piperaquine (DHA-PPQ), artesunate-amodiaquine (ASAQ), and artemether-lumefantrine (AL). The four therapies available in the public sector are chloroquine, amodiaquine, sulphadoxine-pyrimethamine, and AL; sulphadoxine-pyrimethamine (SP) is assumed to have a fixed efficacy of 40%. Two triple ACTs that will be available for deployment in the model are artesunate-mefloquine-piperaquine (ASMQ-PPQ) and ALAQ.

## Pharmacokinetics and pharmacodynamics (PKPD)

As mentioned above, 64 genotypes are modeled reflecting all combinations of two possible alleles at each of the six loci. This meant we had to generate a list of $6 \times 64 = 384$ EC50 values, representing the pharmacodynamic (PD) properties of a particular drug compound on each parasite genotype (see Supplementary Materials in Nguyen et al.[31]). We assumed drugs have independent parasite-killing activity and pharmacokinetic (PK) dynamics, i.e., each drug's EC50 values and half-life are unaffected by the presence of other drugs. Enrolling 10,000 simulated patients in a therapeutic efficacy study, with initial parasitaemia ranging from 2000 to 200,000 parasites/µl (uniform distribution on the $\log_{10}$ scale of 3.3–5.3), gives a 28-day efficacy of 99.89% for ASMQ-PPQ and 99.44% for ALAQ, consistent with field data[18]–Supplementary Fig. 1.

## Simulation protocol

Each simulation consisted of three stages:

- 10 burn-in years calibrated to each historic scenario during which mutation and within-host selection were not allowed (to allow the model to equilibrate at the appropriate PfPR with no additional drug resistance markers).
- 15 years during which mutation and selection processes are active. The last year of this period marks the end of the historic scenario and is denoted as 'year zero'.
- 10-year long prospective scenarios simulating the introduction of TACTs or continued use of ACTs.

At the start of stage 1, public sector drug use was set to 5% of all treatments and increased linearly over[40] twenty years until plateauing at 80% of all treatments in year 5. We assumed that private providers offer sulphadoxine-pyrimethamine (SP), chloroquine (CQ), amodiaquine (AQ), and artemether-lumefantrine (AL). An illustration of the modeling workflow can be found in Supplementary Fig. 55.

## Outcome measures

Two primary model outcome measures were reported: *pfkelch13* 580Y allele frequency over time, and the population-wide treatment failure (TF) rate (the proportion of treatments that result in 28-day parasitological failure as measured by microscopy). The number of simulations of scenarios resulting in malaria elimination over a ten-year period was also recorded. Elimination in both models was defined as all-age PfPR falling below 0.01%, indicative of infections resulting mostly from imported infections.

## Scenarios evaluated

Twenty-seven historic epidemiological scenarios were evaluated, each assuming unique combinations of all-age malaria prevalence (PfPR) (0.1%, 1%, 10%), treatment coverage (25%, 50%, 75%), and baseline ACT choice (DHA-PPQ, ASAQ, AL). Throughout, we refer to the 1% PfPR and 50% treatment coverage with DHA-PPQ scenario as the standard evaluation scenario. For each historic scenario, we explored three different prospective scenarios: (1) continued ACT use as first-line therapy (FLT); (2) switch to artesunate-mefloquine-piperaquine–ASMQ-PPQ; (3) switch to ALAQ. We analyzed additional scenarios where TACTs were introduced late, introduced gradually, or with the introduction of ASMQ-PPQ at a time of varying genotype frequency of resistance markers for all three components of this TACT.

## Metrics of evolutionary dynamics

In malaria infections, as in all evolutionary biology, the amount of time it takes a drug-resistant allele to establish and reach fixation depends on the size of its fitness advantage. In our context, this is defined by a resistant genotype's ability to recrudesce despite a complete course of ACT or TACT. This is because recrudescence is necessary for resistant parasites to reach transmissible densities in the primary selection event, and thereafter recrudescence is an important driver of spread. These recrudescence rates–or treatment failure rates–differ by genotype and treatment as mentioned in the PKPD section above and determine whether a particular therapy has an advantage over another in slowing or controlling drug resistance. Throughout the paper, we refer to treatment failure rates as the percentage of patients who present with a parasitaemia >10 parasites/µl on day 28 following treatment.

In each simulation, we track relevant epidemiological, clinical, and genetic indicators that may provide insights into the evolutionary dynamics at play for each of the explored settings. Alongside treatment failure, the two additional metrics reported throughout are *Pf* prevalence (*PfPR*)–measured as the proportion of all individuals in the population with a parasitaemia over the limit of microscopy detection–and allelic frequencies for the tracked loci, calculated as the weighted number of parasite-positive individuals carrying genotype X divided by the total number of parasite-positive individuals. The weight for each person describes the fraction of their clonal populations carrying genotype X; e.g., an individual hosting five clonal infections, of which two are caused by genotype X, would be given a weight of 2/5.

## Statistical analyses

For each scenario explored, we ran 100 independent stochastic simulations. We consistently summarize the distributions of outcomes of interest using the median and interquartile ranges. Where relevant, we provide the 5th and 95th percentiles of said distributions. We mostly use median results to extrapolate relevant differences across scenarios as in Figs. 2–4. We also provide more standard statistical metrics to infer the

significance of the observed differences when comparing continued ACT use with the introduction of TACTs (ASMQ-PPQ or ALAQ). Supplementary Figs. 20, 21 show the Mann–Whitney $p$ value for each of the 108 comparisons of the two main outcomes: 580Y frequency and treatment failure rate. These $p$ values overwhelmingly support the rejection of the null hypothesis–that the outcome distributions are identical for ACT use and TACT use. As discussed at length in the main text, a switch to TACTs is generally expected to yield a lower frequency of 580Y as well as lower treatment failure rates. A second statistical evaluation of the same comparators involved the estimation of the independent relative risks of acquiring an infection with a parasite carrying a 580Y allele and failing treatment. For each scenario, we took distribution $\omega$ composed of the simulated 580Y frequencies at year 10 for each of the 100 runs as well as distribution $\theta$ from the equivalent treatment failure rates. Taking the population level 580Y frequency at year 10 as the probability that any infection with a malaria parasite will include the 580Y allele, we can then randomly sample X values from $\omega$ and compared them with a uniformly distributed random number (between 0 and 1) to get a synthetic population of X infected individuals, a subset of which carry the 580Y allele. This allows us to extrapolate a measure of relative risk by comparing the subsets carrying the 580Y allele generated from distributions $\omega$ obtained from continued ACT use (non-exposed or control group); against the equivalent subsets obtained from the TACT switch scenarios (exposed or intervention group). For example, for a treatment coverage of 50% and Pf prevalence of 1%, we aggregate the final 580Y frequency for all 100 runs with both continued ACT use and switch to TACT and:

- Sample 1000 values $F$ of 580Y frequency (with replacement) from the distribution of outcomes resulting from 100 model simulations for each scenario.
- Sample 1000 values $U$ from a uniform distribution between 0 and 1 and compare them against the 1000 values obtained in step 1. If $U_i < F_i$ (where $i$ indicates each sample), then $P_i = 1$, otherwise $P_i = 0$. $P$ is the set of synthetic infected individuals with cardinality 1000 and a sum equal to the number of individuals expected to have been infected with a 580Y allele.
- After obtaining a set $P$ for the continued ACT use scenario ($P_{act}$) and an equivalent set for a switch to TACT scenario ($P_{tact}$), we can calculate the relative risk of having an infection with a 580Y allele in the ACT scenario vs. the TACT scenario–probability of infection with 580Y in the ACT scenario / Probability of infection with 580Y in the TACT scenario–simply by using the sums of the corresponding sets, knowing both sets are composed of 1000 samples.

The same calculations were performed for the treatment failure rate outcome and these results are illustrated through forest plots in Fig. 1.

### Reporting summary

Further information on research design is available in the Nature Portfolio Reporting Summary linked to this article.

## Data availability

A repository with the compiled simulation outputs can be found here: https://github.com/Longterm-deployment-of-TACTs/Longterm-deployment-of-TACTs [41].

## Code availability

Both models' code has been made available at: https://github.com/Longterm-deployment-of-TACTs/Longterm-deployment-of-TACTs [41].

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

## Acknowledgements

This research falls under the auspices of the Development of Triple Artemisinin-based Combination Therapies (DeTACT) project funded by UK Aid and the UK Government's Foreign, Commonwealth, and Development Office. We would like to thank all project collaborators for their valuable contribution to project planning and rollout and/or for all the inputs that eventually led to the hypotheses tested in this manuscript. The authors also acknowledge the following sources of funding: UK Aid and the UK Government's Foreign, Commonwealth, and Development Office (C.A., M.D., A.D., N.J.W.). Wellcome Trust grant 220211 (C.A., M.D., A.D., N.J.W.). National Institutes of Health grant NIAID R01AI153355 (M.F.B., T.D.N., T.N.-A.T.). The Bill and Melinda Gates Foundation INV-005517 grant was awarded to Pennsylvania State University (M.F.B., T.D.N., T.N.-A.T.). Bill and Melinda Gates Foundation grant OPP1193472 (R.A.).

## Author contributions

A.D., C.A., M.D., N.J.W., M.B. and R.A. conceptualized the study. T.D.N., B.G., M.B. and R.A. set-up the simulation protocols. T.D.N. and B.G. performed the necessary model calibrations, ran the simulations and compiled the model outputs. TNAT supported the validation of the pharmacodynamic models for each drug/genotype combination. T.D.N., B.G., M.B. and R.A. conducted the subsequent analyses. M.B. and R.A. wrote the initial draft. All authors reviewed the manuscript draft. All authors read and approved the final manuscript.

## Competing interests

The authors declare no competing interests.
