## [Peer Review File · Nature Communications]

Preventing antimalarial drug resistance with triple artemisinin-based combination therapiesREVIEWER COMMENTS

Reviewer #1 (Remarks to the Author):

Progresses made in malaria control over the last few years are now jeopardized by the emergence and spread of resistance to component of Artemisinin-based combination therapies (ACTs), the recommended first lines treatments of uncomplicated falciparum malaria. Number of clinical studies have demonstrated good tolerability, safety and effectiveness of Triple artemisinin-based combination therapies (TACTs) for the treatment of uncomplicated malaria. TACTs can be considered as a potential replacement of ACTs for the treatment of uncomplicated malaria. Studies focusing on long term benefits of TACTs are of greatest importance. The present manuscript is overall well written , the method is accurate with the use of two different models to predict temporal dynamic and treatment outcome of TACTs in areas with different malaria epidemiology. The data obtained using two different mathematical models are promising with a significant delay of drugs resistance following deployment of TACTs recommending their immediate adoption. The limitations of the study have been addressed by the authors . We strongly recommend publication of this manuscript.

Minor comments:

1. Line : In this study, six drug resistance loci were included in the models but the results only focused on mutation at C580Y. Could the authors explain why only projection of 580Y is shown?
2. In their models, the authors project the frequency of the mutation pfk13 580Y which is not seen yet in Africa. Could the authors confirm that the same trend of frequency is suitable for other mutations such as R561H, C499Y or A675V already seen in Africa ?
3. could the authors state whether both models used in this study took account the contribution of other factors such the use of A. Annua as herbal antimalarial in the projection of mutations frequency?

Reviewer #2 (Remarks to the Author):

The authors present a compelling, well-written original research article demonstrating both that two independent models predict triple ACTs slows the build-up of resistance towards principal antimalarial treatments, and that delaying implementation of these triple ACTs can substantially harm disease elimination prospects. The work is a great biological significance to the field. It is relevant to treatment policies in regions where Plasmodium falciparum is yet to establish significant levels of artemisinin resistance. The introduction is sufficiently detailed and comprehensive; they outline their problem nicely. All methodologies (and input parameters) are well-justified, sound and meet the expected standards of the field. Their results are clearly presented and support all conclusions claimed. I would thus be more than delighted to recommend this article for publication in Nature Communications.

I have a few of minor comments that could improve the readability, interpretability and reproducibility of their findings:

1. The high level of complexity to both mathematical models can make interpretation of these results challenging. I do not critique the models themselves as they have been already reviewed, published and deposited on an open-source platform. However, in addition to supplementary table 1, I feel it would be very useful to have a schematic (even if supplementary) of the modelling and analysis workflow used within the paper. This would make understanding the models more accessible, and as a result, interpreting the results in their given contexts would be easier.
2. It is difficult to judge if there is enough detail in the methods for the work to be reproduced since the previously published models that are used are quite complex. It would certainly not be reproducible without looking at the texts and documentation of these original models and their respective code. The previously published models are accessible, and any modifications or extensions to these models have been stated. It would be very useful for the authors to upload their model and analysis code to an open-access resource with appropriate detailed documentation. This would greatly aid in the reproducibility of their work.

3. In Figure 4, it would be good if the median trendlines were removed or made slimmer/dashed, as the piecewise linear interpolation of the median can be a bit misleading, particularly for 580Y allele frequency for low Pf prevalence. The boxplots are sufficient to show the trends.

4. There are a couple of very minor typos:

a. on page 9, line 173, I believe this should be referring to **supplementary** figures 37–39.

b. delete 'and' on page 16, line 345.

Reply to reviewers' comments, submission NCOMMS-23-03328

" Preventing antimalarial drug resistance with triple artemisinin-based combination therapies"

Author's note:

We appreciate the reviewers' constructive comments and feel the revised manuscript should clarify all the concerns raised and be a lot clearer to a wider audience. Particularly, we have made clarifying improvements to several figures, and in the first two paragraphs of the results section as these appear before the methods section (in a Nat Commun article).

All new sources of data and code used to generate the results in the detailed response to reviewer's comments below can be found in the following github repository: <https://github.com/Longterm-deployment-of-TACTs>.

Finally, the science around piperaquine resistance has seen substantial improvements in understanding over the past year. We have added a few references and some additional details in the methods sections on how we model piperaquine resistance.

On behalf of all authors,

Ricardo Aguas

Reviewer 1

1.1. Progresses made in malaria control over the last few years are now jeopardized by the emergence and spread of resistance to component of Artemisinin-based combination therapies (ACTs), the recommended first lines treatments of uncomplicated falciparum malaria. Number of clinical studies have demonstrated good tolerability, safety and effectiveness of Triple artemisinin-based combination therapies (TACTs) for the treatment of uncomplicated malaria. TACTs can be considered as a potential replacement of ACTs for the treatment of uncomplicated malaria. Studies focusing on long term benefits of TACTs are of greatest importance. The present manuscript is overall well written, the method is accurate with the use of two different models to predict temporal dynamic and treatment outcome of TACTs in areas with different malaria epidemiology. The data obtained using two different mathematical models are promising with a significative delay of drugs resistance following deployment of TACTs recommending their immediate adoption. The limitations of the study have been addressed by the authors. We strongly recommend publication of this manuscript.

Author response: We appreciate the reviewer's positive feedback and are very happy to respond to their comments.

1.2. Line : In this study, six drug resistance loci were included in the models but the results only focused on mutation at C580Y. Could the authors explain why only projection of 580Y is shown?

Author response: Yes, this is an excellent comment. To keep the manuscript brief and focused, we presented all results on the endpoints of "580Y frequency" and "treatment failure percentage" only. The referee is right that there are differences in how selection operates on the other resistance loci, depending on which scenario we are looking at.

For example, in Supplementary Figure 34, we see piperazine resistance and artemisinin resistance are co-selected strongly in a scenario where DHA-PPQ is the recommended ACT. The reason is that the treatment failure rate of DHA-PPQ on the double-resistant genotype is very high.

In most scenarios the 76T, 86Y, and 184F alleles are not selected for strongly, as ALAQ puts opposing evolutionary pressures on these alleles (Supplementary Figs 34-36). Continued AL

deployment, on the other hand, facilitates *mdr1* second copy selection (Supplementary Fig 35), which is further enhanced if changing to ASMQ-PPQ but mitigated if changing to ALAQ. AL is the only regimen to strongly select for 84N, although this can be reversed after a policy change to TACTs (Supplementary Fig 35). Under high selection pressure scenarios, ASMQ-PPQ consistently selects for higher frequency of drug-resistance related alleles relative to ALAQ (Supplementary Figs. 34-36). Continued use of ASAQ is predicted to result in high long term 580Y frequencies but barely unnoticeable increases in all other resistance related allele frequencies. We have added a paragraph in the main text referring the reader to these figures.

1.3. In their models, the authors project the frequency of the mutation *pfk13* 580Y which is not seen yet in Africa. Could the authors confirm that the same trend of frequency is suitable for other mutations such as R561H, C499Y or A675V already seen in Africa?

Author response: Yes, good question. The clearance half-lives for R561H and A675V are similar to those of C580Y (see Figure 4 in BMC Medicine paper reviewing these phenotypes ¹). Thus, all other things being equal, these alleles should show the same rates of spread as are seen in our models.

In a manuscript currently under revision (medrxiv, Zupko et al ²) some of the authors have specifically evaluated the spread of R561H in a spatially parameterized Rwandan setting. There is no guarantee that all the spread/invasion rates of artemisinin-resistant alleles are going to be the same, so it will be important to update future TACT analyses with more detailed phenotype information (on the resistant genotypes themselves) and specific parameterizations to African epidemiological scenarios where these alleles have emerged.

Some of these results are in the Zupko et al manuscript references above, but new analyses will need to be done as the resistance situation changes in Africa.

1.4. Could the authors state whether both models used in this study took account the contribution of other factors such the use of *A. Annua* as herbal antimalarial in the projection

¹ <https://bmcmmedicine.biomedcentral.com/articles/10.1186/s12916-018-1207-3>

² <https://www.medrxiv.org/content/10.1101/2022.12.12.22283369v1>

of mutations frequency?

Author response: This is a very interesting question. We realise that in real world contexts, there are various traditional medicines used as antimalarials. We appreciate that the wisdom (which in this case is akin to cross-generational empirical evidence) surrounding the use of herbs such as *A. annua* warrants merit and discussion. However, scientific evidence on the true efficacy of such therapies is lacking. Initial efforts into the pharmacological evaluation of *A. annua* revealed that Artemisinin is only 1 of 29 sesquiterpenes in *A. annua*, some of which show up in much greater concentrations than artemisinin in wild strains of the plant³. In addition, *A. annua* produces at least 36 flavonoids, many of which have antimalarial activity. Although the PKPD dynamics are unclear, “the antimalarial properties of the traditional preparation of *A. annua* most probably reside in the combination of many constituents, not just artemisinin”⁴. Interestingly, the callus of the plant has some antimalarial activity even though it contains no artemisinin⁵. A randomised clinical trial suggested that a 7-day course regimen of *A. annua* has a parasitological cure rate at day 7 of 74% (compared with 91% for quinine in the same trial) alongside relatively high recrudescence rates⁶. The later were as high as 63% at day 28 post treatment which is extremely high and considerably worse than what we would consider realistic for artemether monotherapy. For the reasons listed above, but more importantly, because we cannot predict how 580Y mutations would increase *A. annua* recrudescence rates, we do not include these ethnobotanical therapies in the model.

Reviewer 2

2.1 The authors present a compelling, well-written original research article demonstrating both that two independent models predict triple ACTs slows the build-up of resistance towards principal antimalarial treatments, and that delaying implementation of these triple ACTs can substantially harm disease elimination prospects. The work is a great biological significance to the field. It is relevant to treatment policies in regions where Plasmodium falciparum is yet to establish significant levels of artemisinin resistance. The introduction is sufficiently detailed and comprehensive; they outline their problem nicely. All methodologies (and input parameters) are well-justified, sound and meet the expected

³ Bodeker, Gerard & Willcox, Merlin. (2004). Artemisia annua as a traditional herbal antimalarial.

⁴ Bodeker, Gerard & Willcox, Merlin. (2004). Artemisia annua as a traditional herbal antimalarial.

⁵ François G, Dochez C, Jaziri M, Laurent A. Antiplasmodial activities of sesquiterpene lactones and other compounds in organic extracts of Artemisia annua. Planta Med 1993; 59(suppl):A677–A678.

⁶ Muller, et al. Randomized controlled trial of a traditional preparation of Artemisia annua L. (Annual Wormwood) in the treatment of malaria. Transactions of the Royal Society of Tropical Medicine and Hygiene (2004) 98, 318–321.

standards of the field. Their results are clearly presented and support all conclusions claimed. I would thus be more than delighted to recommend this article for publication in Nature Communications.

Author response: We appreciate the reviewer's positive feedback and are very happy to respond to their comments.

2.2 The high level of complexity to both mathematical models can make interpretation of these results challenging. I do not critique the models themselves as they have been already reviewed, published, and deposited on an open-source platform. However, in addition to supplementary table 1, I feel it would be very useful to have a schematic (even if supplementary) of the modelling and analysis workflow used within the paper. This would make understanding the models more accessible, and as a result, interpreting the results in their given contexts would be easier.

Author response: We appreciate this suggestion and have added a supplementary figure (Supplementary Fig. 55) that we hope will aid comprehension for a wider audience. This figure shows how the models were brought to equilibrium, how mutation rates were aligned so that neither model produced more mutants than the other, how the 'high ACT use period' from 2005-2010 to today was mimicked, and when the policy change (to take place in 2024 or later) was implemented.

2.3 It is difficult to judge if there is enough detail in the methods for the work to be reproduced since the previously published models that are used are quite complex. It would certainly not be reproducible without looking at the texts and documentation of these original models and their respective code. The previously published models are accessible, and any modifications or extensions to these models have been stated. It would be very useful for the authors to upload their model and analysis code to an open-access resource with appropriate detailed documentation. This would greatly aid in the reproducibility of their work.

Author response: Yes, thanks. The model code for both models is open access in GitHub repository/repositories. The GitHub repo(s) contain source code for both models, the analytical pipelines used to generate all results, and the source code used to make all the figures: <https://github.com/Longterm-deployment-of-TACTs>.

2.4 3. In Figure 4, it would be good if the median trendlines were removed or made slimmer/dashed, as the piecewise linear interpolation of the median can be a bit misleading, particularly for 580Y allele frequency for low Pf prevalence. The boxplots are sufficient to show the trends.

Author response: As suggested, we have removed the trendlines.

*2.5 There are a couple of very minor typos: a) on page 9, line 173, I believe this should be referring to *supplementary* figures 37-39. b) delete 'and' on page 16, line 345*

Author response: We thank the reviewer for pointing out these typos. We have made the necessary corrections.

REVIEWERS' COMMENTS

Reviewer #1 (Remarks to the Author):

I am happy with the feedback provided by the authors and do not have any further comment/question. Congratulations!

Reviewer #2 (Remarks to the Author):

I would like to thank the authors for their careful and considerate response to reviewer's comments. All of my previous concerns have been addressed. I am more than happy to recommend the article for publication in its current state.